Mimicry between adult rove beetles and assassin bug nymphs with unequal defenses: antagonistic or mutualistic?

Sugiura Shinji ssugiura@people.kobe-u.ac.jp sugiura.shinji@gmail.com 1
Hayashi Masakazu 2
1 Graduate School of Agricultural Science, Kobe University , Kobe , Hyogo , Japan
2 Hoshizaki Green Foundation , Izumo , Shimane , Japan
Brygadyrenko Viktor
Electronic publication date: 2025 Sep 9
Publication date: 2025
Volume: 13
Electronic Location ID: e19942
Received 2025 Apr 19; Accepted 2025 Jul 28
Copyright: ©2025 Sugiura and Hayashi
Copyright year: 2025
Copyright holder: Sugiura and Hayashi
License: This is an open access article distributed under the terms of the Creative Commons Attribution License, which permits unrestricted use, distribution, reproduction and adaptation in any medium and for any purpose provided that it is properly attributed. For attribution, the original author(s), title, publication source (PeerJ) and either DOI or URL of the article must be cited.
License URL: https://creativecommons.org/licenses/by/4.0/

Keywords: Chemical defences, Co-mimics, Frogs, Mimicry rings, Müllerian mimicry, Quasi-Batesian mimicry, Reduviidae, Staphylinidae

Funding: Grants-in-Aid for Scientific Research (JSPS KAKENHI) JP19K06073 JP24K02099 Hoshizaki Green Foundation This study was supported by the Grants-in-Aid for Scientific Research (JSPS KAKENHI Grant numbers JP19K06073 and JP24K02099), as well as funds from the Hoshizaki Green Foundation. The funders had no role in study design, data collection and analysis, decision to publish, or preparation of the manuscript.

==============================
Defensive mimicry encompasses a continuum ranging from Batesian to Müllerian mimicry. Batesian mimicry involves antagonistic interactions between undefended and defended species, whereas Müllerian mimicry represents mutualistic interactions between species with comparable levels of defense. When mimicry occurs between species with unequal defensive abilities, it is termed quasi-Batesian mimicry, though whether such interactions are antagonistic or mutualistic remains debated. Despite their common occurrence in nature, few quasi-Batesian mimicry systems have been experimentally studied. Here, we investigated the mimetic interaction between two chemically defended insect species, the rove beetle Paederus fuscipes Curtis, 1826 (Coleoptera: Staphylinidae) and the assassin bug Sirthenea flavipes (Stål, 1855) (Hemiptera: Reduviidae), through behavioral assays with their potential predator, the pond frog Pelophylax nigromaculatus (Hallowell, 1861) (Anura: Ranidae), which naturally co-occurs with these insects in Japan. Adult P. fuscipes resemble S. flavipes nymphs in their conspicuous reddish-orange and black coloration. Under laboratory conditions, 45.8% of pond frogs rejected P. fuscipes adults, whereas 70.8% rejected S. flavipes nymphs, suggesting that the assassin bug nymphs are better defended. Prior exposure to S. flavipes increased frog rejection of P. fuscipes, whereas exposure to P. fuscipes slightly reduced rejection of S. flavipes. These results indicate that adult P. fuscipes may gain protective benefits from mimicry of S. flavipes nymphs, while the latter may incur a small cost.

Introduction

Many animals, particularly invertebrates, possess defensive chemicals that help deter predators (Eisner, Eisner & Siegler, 2005; Sugiura, 2020a). In many cases, these chemically defended species also display conspicuous body color patterns that serve as warning (aposematic) signals to potential predators (Quicke, 2017; Ruxton et al., 2018). Aposematic species frequently form mimicry rings involving various types of mimicry (Kunte, Kizhakke & Nawge, 2021). Mimetic interactions are generally categorized into two types: Batesian mimicry, in which undefended species mimic defended species (Bates, 1862; Ruxton et al., 2018), and Müllerian mimicry, in which equally defended species share warning signals (Müller, 1878; Müller, 1879; Sherratt, 2008; Ruxton et al., 2018). Defensive mimicry can thus be seen as a continuum, with Batesian and Müllerian mimicry at its extremes (Balogh, Gamberale-Stille & Leimar, 2008). Within this continuum, interactions involving unequally defended species—commonly referred to as quasi-Batesian mimicry (Speed, 1993; Speed, 1999; Speed & Turner, 1999; Rowland et al., 2010)—are influenced not only by differences in defensive strength but also by predator learning behavior, signal similarity, and sampling strategies (Sherratt, 2008; Aubier, Joron & Sherratt, 2017). Whether these interactions are ultimately antagonistic or mutualistic may thus depend on how predators generalize or discriminate between signals during foraging and learning.

Mimetic interactions involving species with unequal levels of defense are common in nature (e.g., Marples, Brakefield & Cowie, 1989; Marples, 1993; Quicke, 2017; Winters et al., 2018; Chouteau et al., 2019; Soukupová, Veselý & Fuchs, 2021). However, whether such interactions are ultimately antagonistic or mutualistic remains controversial (Speed et al., 2000; Rowland et al., 2007; Rowland et al., 2010; Sugiura & Hayashi, 2023). Some experimental studies using artificial prey and avian predators suggest that quasi-Batesian mimicry is antagonistic (Speed et al., 2000; Rowland et al., 2010), whereas others found no such evidence (Lindström et al., 2006; Rowland et al., 2007). In contrast, studies involving real co-mimics and their shared predators have demonstrated mutualistic outcomes (Raška et al., 2020; Sugiura & Hayashi, 2023). Despite their potential ecological importance, mimetic interactions involving unequally defended species have been experimentally studied in only a limited number of studies (Pekár et al., 2017; Raška et al., 2020; Sugiura & Hayashi, 2023; Pekár et al., 2024).

Rove beetles of the genus Paederus Fabricius, 1775 (Coleoptera: Staphylinidae) produce a potent hemolymph toxin known as pederin, which is synthesized by endosymbiotic bacteria (Kellner, 2002; Piel, 2002). Contact with pederin-laden hemolymph causes severe dermatitis in humans, termed dermatitis linearis (Frank & Kanamitsu, 1987; Borroni et al., 1991; Oliver & Perlman, 2020). Unlike other hemolymph-toxic beetles (e.g., Coccinellidae, Meloidae, Oedemeridae, Lampyridae), Paederus beetles do not reflex bleed; instead, hemolymph is released only when they are injured (Dettner, 1987). Pederin, a highly cytotoxic compound, has been shown to protect both larvae and adults of Paederus from predators such as wolf spiders (Kellner & Dettner, 1996) and carabid beetles (Tabadkani & Nozari, 2014). Several Paederus species exhibit conspicuous reddish-orange and black coloration, believed to serve as aposematic warning signals (Dettner, 1987; Tabadkani & Nozari, 2014; Parker, 2017). Paederus fuscipes Curtis, 1826 (Fig. 1A), one of the most widespread species in the genus, is distributed across Asia and Europe and has been implicated in dermatitis outbreaks worldwide (Frank & Kanamitsu, 1987; Gao et al., 2024). Adults are commonly observed from May to October in grassland and farmland habitats in central Japan, and under laboratory conditions they feed on small invertebrates as well as fresh animal and plant tissues (Kurosa, 1958). The hemolymph of P. fuscipes contains pederin, which is present throughout all developmental stages from eggs to adults (Kellner & Dettner, 1995). Female adults contain higher levels of pederin than males (Kellner & Dettner, 1995). Adults of P. fuscipes have a striking reddish-orange and black coloration (Fig. 1A), which closely resembles that of the nymphs of the assassin bug Sirthenea flavipes (Staal, 1855) (Hemiptera: Reduviidae), a species that co-occurs with P. fuscipes in grasslands in Japan (Fig. 1B) (Hayashi, 2023). However, no studies have experimentally evaluated whether this resemblance reflects a true mimetic relationship.

Figure 1 A rove beetle, an assassin bug, and their potential predator.

(A) An adult rove beetle, Paederus fuscipes. (B) A nymph (fourth instar) of the assassin bug, Sirthenea flavipes. (C) A juvenile pond frog, Pelophylax nigromaculatus. These photos were taken at the same site in Shimane Prefecture on 21 July 2023. Photo credit: Shinji Sugiura.

The assassin bug S. flavipes is broadly distributed in Asia (Chłond, 2018). Both nymphs and adults inhabit the ground surface or shallow soil layers in grassland and farmland and are known to prey on the mole cricket Gryllotalpa orientalis Burmeister, 1839 (Orthoptera: Gryllotalpidae) (Hayashi, 2023; Sugiura & Hayashi, 2023). In Shimane Prefecture, central Japan, overwintered adults oviposit between June and August, with nymphs hatching during the same period (Hayashi, 2023). The species undergoes five nymphal instars before adult emergence, which begins around September (Hayashi, 2023). Sirthenea flavipes shares habitats with P. fuscipes and uses its proboscis to inject venom into both prey and predators (Hayashi, 2023; Sugiura & Hayashi, 2023; Sugiura & Hayashi, 2025). In addition, chemicals present on the body surface of S. flavipes deter potential predators (Sugiura & Hayashi, 2023). Because nymphs of S. flavipes exhibit the same reddish-orange and black coloration (Fig. 1B) (Ishikawa, Takai & Yasunaga, 2012), Hayashi (2023) proposed that they may be mimics of adult P. fuscipes. Given that both species are chemically defended, the resemblance may reflect Müllerian or quasi-Batesian mimicry. However, whether this resemblance results in mutualistic outcomes remains unclear.

To investigate the nature of the mimetic interaction between adult P. fuscipes and S. flavipes nymphs, we conducted behavioral assays using a potential shared predator, the pond frog Pelophylax nigromaculatus (Hallowell, 1861) (Anura: Ranidae). Pelophylax nigromaculatus is native to East Asia, including Japan (Matsui & Maeda, 2018). Post-metamorphic juveniles and adults inhabit grasslands surrounding ponds and paddy fields and feed on both aquatic and terrestrial arthropods (Hirai & Matsui, 1999; Hirai, 2002; Sano & Shinohara, 2012; Sarashina, Yoshihisa & Yoshida, 2011). Juvenile pond frogs co-occur with P. fuscipes and S. flavipes in grasslands in Japan (Fig. 1C). Kurosa (1958) recorded P. fuscipes adults in the stomach contents of wild Pelophylax and other frogs, indicating that these frogs are predators of P. fuscipes. Thus, P. nigromaculatus is a suitable model predator for evaluating the defensive effectiveness and potential mimicry between the two insect species. We first compared the rejection rates of P. nigromaculatus toward adult P. fuscipes and S. flavipes nymphs to assess differences in their defensive efficacy. To further test the effects of prior exposure, we examined whether previous encounters with one species altered frog responses toward the other. Finally, we discuss the implications of these results for the structure and dynamics of mimicry rings involving P. fuscipes and S. flavipes.

Materials and Methods

Study species and sampling

Adults of the rove beetle P. fuscipes were collected from grasslands in Hyogo and Shimane Prefectures (Honshu, Japan) in June–August 2022 and in July–August 2023. Beetles were maintained in plastic containers (100 mm diameter × 100 mm height) at 25 °C and were fed insect jelly (Pro Jelly; KB Farm, Koreosu Co. Ltd., Saitama, Japan). Prior to experiments, body length and weight were measured to the nearest 0.01 mm and 0.1 mg using digital calipers (CD-15AX, Mitutoyo, Kawasaki, Japan) and an electronic balance (CPA64, Sartorius Japan K.K., Tokyo, Japan), respectively (Table 1). Sex was identified based on abdominal morphology under a stereomicroscope (Kurosa, 1958). A total of 34 adults (19 males, 15 females) were used in subsequent experiments.

Table 1 Body sizes of rove beetles, assassin bugs, and pond frogs used in this study.

Species	Rove beetle	Assassin bug	Pond frog	
	Paederus fuscipes	Sirthenea flavipes	Pelophylax nigromaculatus	
Stage	Adult	First–fourth instar	Juvenile–subadult	
Body length mma	8.0 ± 0.1 (7.1–8.9)	6.7 ± 0.3 (3.5–9.9)	31.7 ± 0.9 (20.7–42.0)	
Body weight mga	3.8 ± 0.1 (2.4–5.5)	9.0 ± 1.3 (1.7–25.3)	2,572.6 ± 190.6 (702.9–5,641.4)	
n	34	29	48	
Notes.

a Mean ± standard errors (range: minimum–maximum).

Nymphs of the assassin bug S. flavipes were reared from eggs laid by several females collected in Shimane during July–August 2022 and 2023. Additional nymphs were collected from a grassland site in Shimane in July 2023. Nymphs were housed in plastic containers (85 mm diameter × 25 mm height) under laboratory conditions (25 °C) and were fed G. orientalis nymphs (Hayashi, 2023). Body length and weight were measured using the same instruments as above (Table 1), and developmental instars were determined based on the number of molts and body size (Hayashi, 2023). A total of 29 nymphs (first–fourth instars) were used for the experiments.

Post-metamorphic juveniles of the pond frog P. nigromaculatus were collected from Hyogo Prefecture in June–August 2022 and in July–August 2023. Frogs were individually housed in plastic cages (120 mm × 85 mm × 130 mm) under laboratory conditions (25 °C), and were fed live mealworms [larvae of Tenebrio molitor Linnaeus, 1758 (Coleoptera: Tenebrionidae)] and nymphs or adults of the cockroach Periplaneta lateralis Walker, 1868 (Blattodea: Blattidae) (Sugiura, 2018; Sugiura, 2020b; Sugiura & Tsujii, 2022). Frogs that easily fed on mealworms or cockroaches were used for behavioral experiments. Snout–vent length and body weight were recorded to the nearest 0.01 mm and 0.1 mg, respectively (Table 1). A total of 48 juveniles and subadults were used in this study.

Experiment I: initial response tests

To assess whether adult P. fuscipes or S. flavipes nymphs are better defended against pond frogs, we conducted behavioral assays under well-lit laboratory conditions (25 °C) at Kobe University in August 2022 and 2023, following the protocol of Sugiura & Hayashi (2023). Each frog was placed individually into a plastic cage (120 mm × 85 mm × 130 mm). After an acclimation period, a single adult P. fuscipes or S. flavipes nymph was introduced into the cage (Figs. 2A and 2B). Frogs were food-deprived for at least 24 h prior to the experiment to standardize hunger levels (Honma, Oku & Nishida, 2006; Sugiura, 2020b; Sugiura & Hayashi, 2024). Frog and insect behaviors were recorded using a digital camera (iPhone 12 Pro Max; Apple Inc., Cupertino, CA, USA) and a digital video camera (Handycam HDR-PJ790V, Sony, Japan). We carefully reviewed the video footage using QuickTime Player ver. 10.5 to assess how each frog responded to the presented insect and how each insect defended itself. If a frog did not initiate an attack on the insect, we classified the response as “ignore”. If a frog attacked the insect but subsequently ceased its attack, we classified the response as “stop attack”. If the frog swallowed the insect, we classified this outcome as “eat”. We interpreted both “ignore” and “stop attack” as rejection responses. Insect species were assigned based on availability at the time of each trial, rather than formal randomization. However, the sequence of species presentation was not systematically ordered and appeared approximately random across trials. A post hoc analysis confirmed that presentation order did not affect predation outcomes. In total, 24 adult P. fuscipes (12 males and 12 females), 24 S. flavipes nymphs, and 48 frogs were used. No individual insect or frog was used more than once in this experiment. The body sizes of frogs that attacked P. fuscipes did not significantly differ from those that attacked S. flavipes (Welch’s t-test; snout–vent length: t = 0.006, df = 45.10, P = 0.995; body weight: t = −0.088, df = 45.49, P = 0.931). The sample size was determined based on the number of S. flavipes nymphs available for testing. The same frogs used in Experiment I were later used in generalization tests (Experiment II).

Figure 2 Experimental procedures and summary of results.

(A) Experiment I (initial response test): an adult rove beetle (Paederus fuscipes) was provided to a pond frog (Pelophylax nigromaculatus). (B) Experiment I: an assassin bug nymph (Sirthenea flavipes) was provided to a frog. (C) Experiment II (generalization test): an assassin bug nymph was provided to the frog that had encountered the rove beetle. (D) Experiment II: an adult rove beetle was provided to the frog that had encountered the assassin bug. Photo credit: Shinji Sugiura.

Experiment II: generalization tests

To evaluate whether prior experience with one insect species influences the frog’s response to the other species, we conducted generalization tests using the same frog individuals from Experiment I. In one treatment, each frog was first presented with an adult P. fuscipes (i.e., Experiment I; Fig. 2A), and then, approximately 6 min later (median = 6 min; range = 5–7 min), the same frog was presented with a S. flavipes nymph (Fig. 2C; n = 24). In the other treatment, each frog was first presented with a S. flavipes nymph (i.e., Experiment I; Fig. 2B), and then, approximately 6 min later (median = 6 min; range = 5–11 min), it was presented with an adult P. fuscipes (Fig. 2D; n = 24). The order of species presentation was fully balanced across individuals (24 frogs per treatment) to control for potential order effects. The interval between exposures (median = 6 min; range = 5–11 min) was selected based on a previous study using P. nigromaculatus in the similar generalization tests (Sugiura & Hayashi, 2023), and was intended to capture short-term behavioral adjustments. Given that both the predator species and experimental paradigm were consistent with those of the previous study, we considered this interval appropriate for detecting short-term generalization responses. Although this interval is shorter than the generalization or memory durations reported for other predators—such as spiders tested 50 min and 24 h after exposure (Raška et al., 2020), and birds retaining avoidance for 1–4 weeks (Kojima & Yamamoto, 2020)—it was sufficient to test immediate associative responses.

Frog and insect behaviors were recorded using the same cameras described above, and the video footage was carefully reviewed to assess the responses of each frog and insect. To rule out satiation as a cause for prey rejection, we offered a palatable prey item (mealworm) to each frog that rejected either insect in Experiment II (Sugiura & Sato, 2018; Sugiura, 2018). In total, 11 adult P. fuscipes (eight males, three females), 16 S. flavipes nymphs, and 48 frogs were used in Experiment II. Unlike in Experiment I, some insects were reused in Experiment II. However, no frog was tested more than once in a given treatment.

All experimental procedures were conducted in accordance with the Animal Experimentation Regulations of Kobe University (Nos. 30–01 and 2023–03). After the experiments, the frogs were maintained in our laboratory for use in other studies (e.g., Sugiura & Hayashi, 2024).

Data analysis

All statistical analyses were conducted using R version 4.4.1, with a significance threshold set at α = 0.05.

We used Welch’s t-tests to compare (1) the body size (length and weight) of adult rove beetles and assassin bug nymphs used in Experiment I, and (2) the body size (snout–vent length and body weight) of pond frogs that attacked either insect species in the same experiment.

To assess the effects of insect species, insect size, and frog size on predation outcomes, we constructed generalized linear models (GLMs) with a binomial distribution and a logit link function. The binary response variable was frog behavior: successful predation (1) or rejection (0), with rejection defined as either ignoring or ceasing to attack the insect. Explanatory variables included insect species (P. fuscipes or S. flavipes), insect body size (length or weight), and frog body size (snout–vent length or weight). In cases where the residual deviance was substantially greater than the residual degrees of freedom, indicating overdispersion, we employed a quasi-binomial distribution instead of the binomial one.

To evaluate how prior experience influenced frog responses to insects, we fitted a generalized linear mixed model (GLMM) with a binomial distribution and logit link using the glmmTMB package version 1.1.10. The binary response variable again reflected predation outcome (1 = predation, 0 = rejection). Explanatory variables included insect species, encounter history (initial exposure or after exposure to the other species), and their interaction. Random effects accounted for repeated use of individual insects and frogs in Experiment II, thereby controlling for potential variation associated with insect reuse and inter-individual behavioral differences among frogs. When the interaction between insect species and frog encounter history was not significant, we used estimated marginal means (EMMs) derived from the GLMM with a binomial distribution. EMMs were calculated using the emmeans package version 1.10.7. This analysis allowed us to interpret the estimated predation probabilities for each combination of insect species and frog encounter history, taking into account the explanatory variables and random effects in the model.

For all GLM, GLMM, and EMM analyses, odds ratios (ORs) and their 95% confidence intervals (CIs) were calculated by exponentiating model coefficients and standard errors. For EMM contrasts, ORs and CIs were computed using the emmeans package.

Multicollinearity was assessed using variance inflation factors (VIFs) for both the GLMs and the GLMM. All values fell within commonly accepted thresholds, indicating that multicollinearity was unlikely to have influenced model estimates. Potential overfitting was assessed by calculating the ratio of observations to the number of model parameters. In both models, the ratio exceeded the commonly used threshold of 10, suggesting that the models were not overfitted. For the GLMM, residual diagnostics using the DHARMa package version 0.4.7 indicated no substantial deviation from model assumptions, suggesting adequate model fit.

Results

Experiment I: initial response tests

Of the 24 pond frogs presented with adult rove beetles, 21 individuals (87.5%) attacked the beetles (Fig. 2A), while three (12.5%) ignored them (Table 2). Thirteen frogs (54.2%) ate the beetles (Fig. 2A), whereas eight frogs (33.3%) stopped attacking within 2 s after their tongues made contact (Fig. 3A; Table 2; Video S1). The predation rate on male beetles (58.3%) was not substantially different from that on female beetles (50.0%; Table 2).

Table 2 Behavioral responses of pond frogs to adult rove beetles and assassin bug nymphs in Experiment I.

Insect species	Stage	Frog response % (n)a	
		Eat	Stop attack	Ignore	Total	
Rove beetle (Paederus fuscipes)	Female adult	50.0 (6)	41.7 (5)	8.3 (1)	100.0 (12)	
	Male adult	58.3 (7)	25.0 (3)	16.7 (2)	100.0 (12)	
	Total	54.2 (13)	33.3 (8)	12.5 (3)	100.0 (24)	
Assassin bug (Sirthenea flavipes)	Nymph	29.2 (7)	62.5 (15)	8.3 (2)	100.0 (24)	
Notes.

a “Eat”: pond frogs successfully consumed adult rove beetles (or assassin bug nymphs). “Stop attack”: frogs released adult rove beetles (or assassin bug nymphs) after their tongues contacted them. “Ignore”: frogs did not attack adult rove beetles (or assassin bug nymphs).

Figure 3 Pond frog rejecting an adult rove beetle and an assassin bug nymph.

(A) A pond frog (Pelophylax nigromaculatus) stopped attacking an adult rove beetle (Paederus fuscipes) immediately after its tongue contacted it. (B) A pond frog stopped attacking an assassin bug nymph (Sirthenea flavipes) immediately after its tongue contacted it. Credit: Shinji Sugiura.

When presented with assassin bug nymphs, 22 frogs (91.7%) attacked them (Fig. 2B), while two (8.3%) ignored them (Table 2). Seven frogs (29.2%) ate the nymphs (Fig. 2B), and 15 (62.5%) stopped attacking within 1 s after tongue contact (Fig. 3B; Table 2; Video S2). Assassin bug nymphs rarely attempted to stab frogs with their proboscises during frog attacks.

All rove beetles and assassin bugs that were either ignored or released after initial attack remained alive. The predation success rate was higher for rove beetles (54.2%) than for assassin bug nymphs (29.2%; Table 2). The two insect species differed significantly in body size: adult rove beetles were longer but lighter than assassin bug nymphs (Table 1; Welch’s t-test, body length: t = 3.840, df = 28.253, P = 0.0006; body weight: t = −3.812, df = 23.66, P = 0.0009). However, GLM analysis showed no significant effects of insect species, insect body size, or frog body size on predation success (Table 3). The odds ratio (OR) for predation on S. flavipes relative to P. fuscipes was 0.457 (95% CI [0.111–1.878]; Table 3A) or 0.287 (0.065–1.266; Table 3B).

Table 3 Results of generalized linear models (GLMs) identifying factors that influenced the predation success of pond frogs on adult rove beetles and assassin bug nymphs in Experiment I.

(A) Effects of insect species, body length, and frog snout-vent length.	
Response variable	Explanatory variable	Coefficient estimate	Standard error	t	OR	95% CI for OR (lower–upper)	P	
Predation successa	Intercept	−1.693411	2.613384	−0.648	0.184	0.001–30.840	0.520	
	Insect species (vs. assassin bugs)b	−0.783593	0.721278	−1.086	0.457	0.111–1.878	0.283	
	Insect body length	0.241601	0.322645	0.749	1.273	0.677–2.396	0.458	
	Frog snout–vent length	−0.002508	0.057717	−0.043	0.997	0.891–1.117	0.966	
(B) Effects of insect species, body weight, and frog weight.	
Response variable	Explanatory variable	Coefficient estimate	Standard error	t	OR	95% CI for OR (lower–upper)	P	
Predation successa	Intercept	−0.02745	0.7509	−0.037	0.973	0.223–4.239	0.971	
	Insect species (vs. assassin bugs)b	−1.249	0.7577	−1.649	0.287	0.065–1.266	0.106	
	Insect weight	0.03296	0.06795	0.485	1.034	0.905–1.181	0.630	
	Frog weight	0.00002622	0.0002524	0.104	1.000	1.000–1.001	0.918	
Notes.

a A quasi-binomial distribution was used instead of the binomial distribution because the residual deviance was substantially larger than the residual degrees of freedom.

b Adult rove beetles were used as a reference.

Experiment II: generalization tests

Frogs previously exposed to assassin bug nymphs were subsequently presented with adult rove beetles (n = 24; Fig. 2D; Table 4A). Of these, 10 frogs (41.7%) attacked the beetles (Fig. 2D), while 14 (58.3%) ignored them (Table 4A; Fig. 4A). Five frogs (20.8%) ate the beetles (Fig. 2D). Nearly all frogs that ignored the beetles accepted mealworms afterward (Table 4A), indicating that their responses were not due to satiation.

Table 4 Results of generalization tests: behavioral responses of pond frogs to adult rove beetles and assassin bug nymphs after encountering the other insect species.

(A) Responses of pond frogs to assassin bug nymphs in Experiment I and adult rove beetles in Experiment II.	
		Experiment II: frog response to rove beetle	
	Frog behaviora	Eat	Stop attack	Ignore	Total	
Experiment I: frog response to assassin bug	Eat	2	1	4b	7	
	Stop attack	3	3	9	15	
	Ignore	0	1	1	2	
	Total	5	5	14	24	
(B) Responses of pond frogs to adult rove beetles in Experiment I and assassin bug nymphs in Experiment II.	
		Experiment II: frog response to assassin bug	
	Frog behaviora	Eat	Stop attack	Ignore	Total	
Experiment I: frog response to rove beetle	Eat	8	5	0	13	
	Stop attack	0	4	4	8	
	Ignore	0	0	3	3	
	Total	8	9	7	24	
Notes.

a “Eat”: pond frogs successfully consumed adult rove beetles (or assassin bug nymphs). “Stop attack”: frogs released adult rove beetles (or assassin bug nymphs) after their tongues contacted them. “Ignore”: frogs did not attack adult rove beetles (or assassin bug nymphs).

b Only one frog did not consume a mealworm after Experiment II. All other frogs that had ignored rove beetles or assassin bugs in Experiment II ate mealworms afterward.

Figure 4 Behavioral responses of pond frogs to adult rove beetles and assassin bug nymphs in Experiments I and II.

(A) Responses of pond frogs (Pelophylax nigromaculatus) to adult rove beetles (Paederus fuscipes). (B) Responses of pond frogs to assassin bug nymphs (Sirthenea flavipes). “Exp. I”: initial response tests (Experiment I). “Exp. II”: generalization tests (Experiment II). “Eat”: pond frogs successfully consumed adult rove beetles or assassin bug nymphs. ”Stop attack”: frogs released rove beetles or assassin bugs after their tongues had contacted them. “Ignore”: frogs did not attack rove beetles or assassin bugs. Photo credit: Shinji Sugiura.

In the reciprocal treatment, frogs that had previously encountered adult rove beetles were presented with assassin bug nymphs (n = 24; Fig. 2C; Table 4B). Seventeen frogs (70.8%) attacked the nymphs (Fig. 2C), and seven (29.2%) ignored them (Table 4B; Fig. 4B). Eight frogs (33.3%) ultimately ate the nymphs (Fig. 2C). All frogs that ignored the bugs accepted mealworms afterward (Table 4B).

Exposure to assassin bug nymphs markedly reduced the attack rate on adult rove beetles from 87.5% (Fig. 2A) to 41.7% (Fig. 2D), and also reduced the predation rate from 54.2% to 20.8% (Fig. 4A). Conversely, prior exposure to adult rove beetles reduced the attack rate on assassin bug nymphs from 91.7% (Fig. 2B) to 70.8% (Fig. 2C), but the predation rate slightly increased from 29.2% to 33.3% (Fig. 4B).

The GLMM analysis revealed that frog encounter history had a significant effect on predation success, whereas insect species and the interaction between insect species and frog encounter history were not statistically significant (Table 5). The odds ratio for the encounter order effect was 0.085 (95% CI [0.008–0.862], P = 0.037; Table 5), while the interaction between prey species and order was not significant (OR = 15.8, 95% CI [0.458–548.0], P = 0.126; Table 5).

Table 5 Results of a generalized linear mixed model (GLMM) identifying factors that influenced the predation success of pond frogs on adult rove beetles and assassin bug nymphs in Experiment II.

Response variable	Explanatory variable	Coefficient estimate	Standard error	z	OR	95% CI for OR (lower–upper)	P	
Predation success	Intercept	0.2893	0.6921	0.418	1.34	0.344–5.19	0.6759	
	Insect species (vs. assassin bugs)a	−1.7992	1.0964	−1.641	0.165	0.019–1.42	0.1008	
	Frog encounter history (vs. encounter)b	−2.4691	1.1842	−2.085	0.085	0.008–0.862	0.0371	
	Insect species × frog encounter history	2.7627	1.8079	1.528	15.8	0.458–548.0	0.1265	
Notes.

a Adult rove beetles were used as a reference.

b Initial responses of frogs were used as a reference.

The EMMs analysis showed that prior encounters with assassin bug nymphs significantly altered frog predation behavior toward rove beetles (OR = 11.81, 95% CI [1.22–114.28], p = 0.037; Table 6), but similar exposure to rove beetles did not affect responses to assassin bug nymphs (OR = 0.75, 95% CI [0.16–3.47], P = 0.771; Table 6).

Table 6 Estimated marginal means (EMMs) of predation probability and odds ratios for the effect of frog encounter history on predation of each insect species.

(A) EMMs of predation probability.	
Insect species	Frog encounter history	Estimated marginal meanc	
		Probability	Standard error	95% CI for probability	
Rove beetle (Paederus fuscipes)	Initiala	0.572	0.169	0.2559–0.838	
	Post-encounterb	0.102	0.0851	0.0179–0.413	
Assassin bug (Sirthenea flavipes)	Initiala	0.181	0.122	0.0420–0.527	
	Post-encounterb	0.229	0.138	0.0600–0.579	
(B) Odds ratios for the effect of frog encounter history on predation success.	
Insect species	Comparison	Standard error	z	Odds ratiod	95% CI for OR	P	
Rove beetle (Paederus fuscipes)	Initialavs. Post-encounterb	14.000	2.085	11.812	1.22–114.28	0.0371	
Assassin bug (Sirthenea flavipes)	Initialavs. Post-encounterb	0.751	−0.292	0.746	0.16–3.47	0.7707	
Notes.

a Initial responses of pond frogs (Pelophylax nigromaculatus) that have not encountered either insect species.

b Responses of pond frogs that have encountered the other insect species.

c Predicted probabilities (mean ± standard errors) were back-transformed from the logit scale.

d Odds ratios and their confidence intervals were calculated on the log odds ratio scale.

Discussion

Whether mimetic interactions between unequally defended species are mutualistic or antagonistic remains a topic of ongoing debate (Speed et al., 2000; Rowland et al., 2007; Rowland et al., 2010; Aubier, Joron & Sherratt, 2017; Sugiura & Hayashi, 2023; Sugiura & Hayashi, 2025). In this study, we examined the nature of the mimetic interaction between adult rove beetles (P. fuscipes) and assassin bug nymphs (S. flavipes) by assessing their interactions with a shared predator, the pond frog P. nigromaculatus. In our initial response tests (Experiment I), P. fuscipes adults were rejected less frequently by frogs than S. flavipes nymphs (Fig. 4), although this difference in defensive efficacy was not statistically significant (Table 3). In the generalization tests (Experiment II), frogs that had previously encountered S. flavipes nymphs showed reduced predation on P. fuscipes adults compared to frogs with no such exposure (Fig. 4A). In contrast, prior experience with P. fuscipes slightly increased the predation rate on S. flavipes nymphs (Fig. 4B). While the interaction between insect species and encounter history was not statistically significant (Table 5), species-specific comparisons showed that prior exposure significantly reduced predation on P. fuscipes but had no such effect on S. flavipes (Table 6). These results suggest a potentially asymmetric mimetic relationship, with P. fuscipes gaining protection from the resemblance, whereas S. flavipes appears to incur minimal cost under the tested conditions.

Pond frogs as predators

The pond frog P. nigromaculatus has been widely used as a model predator to test the effectiveness of anti-predator defenses in a variety of insects (Sugiura, 2020a), including grasshoppers (Honma, Oku & Nishida, 2006), assassin bugs (Sugiura & Hayashi, 2023), wasps (Sugiura & Tsujii, 2022; Sugiura & Urano, 2025), bombardier beetles (Sugiura, 2018; Sugiura & Hayashi, 2023), and aquatic beetles (Sugiura, 2020b; Sugiura & Hayashi, 2024). When presented with chemically defended insects, P. nigromaculatus typically attacks but quickly releases the prey upon tongue contact (Sugiura, 2018; Sugiura & Hayashi, 2023; Sugiura & Hayashi, 2024). A similar pattern was observed in this study when frogs rejected both P. fuscipes adults and S. flavipes nymphs (Fig. 3), suggesting that both species possess chemical defenses that deter frog predation.

In the generalization tests, frogs often ignored the second insect they encountered after an initial interaction with the other species (Fig. 4), implying that they may not visually discriminate between P. fuscipes adults and S. flavipes nymphs. Importantly, these frogs readily consumed palatable mealworms after rejecting either insect, indicating that they distinguished the defended prey from edible alternatives. According to generalization theory (Ruxton et al., 2008), such behavior may arise from the frog’s ability to generalize the warning signals of one defended species to others with similar appearance (Sugiura & Hayashi, 2023; Sugiura & Hayashi, 2025). This pattern of behavior suggests that frogs form internal representations—or signal templates—based on previous encounters with defended prey. Such templates allow predators to classify novel prey items as potentially unprofitable based on visual similarity alone, a process that underlies both the convergence of warning signals and the evolutionary stability of mimicry systems (Ruxton et al., 2008). Therefore, the frogs’ generalization responses observed in our experiments are not merely behavioral reactions, but reflect cognitive processes that are central to mimicry theory. Our findings are consistent with theoretical models predicting that the shape and strength of predator generalization may influence the dynamics of signal evolution in mimicry complexes.

Although previous studies suggest that P. nigromaculatus has relatively short memory retention compared to other vertebrate predators such as birds (Sugiura & Hayashi, 2023), a few frogs in our study refrained from attacking the insects even in the initial response trials (Experiment I; Table 2; Fig. 4). These individuals were collected from locations where P. fuscipes and/or S. flavipes naturally occur, and thus may have had prior experience with these prey species. Such prior exposure could explain the observed avoidance behavior and suggests that memory retention may sometimes exceed the durations typically assumed in laboratory experiments. Consequently, the use of wild-caught frogs and our experimental timing may have influenced the observed outcomes.

Adult rove beetles as mimics

Many adult rove beetles secrete defensive chemicals from their abdomens (Dettner, 1987; Eisner, Eisner & Siegler, 2005). Both male and female adults of P. fuscipes produce secretions from an abdominal gland located at the anterior margin of the fourth sternite (Kellner & Dettner, 1992); these secretions contain higher alkanes, which may act as deterrents to predators (Wen & Ueno, 2022). In addition, P. fuscipes harbors the potent hemolymph toxin pederin, which is produced by endosymbiotic bacteria (Kellner, 2002; Piel, 2002). Pederin disrupts mitosis in eukaryotic cells by inhibiting protein synthesis in ribosomes, and it has been shown to deter lycosid spiders and carabid beetles from preying on P. fuscipes larvae and adults, respectively (Kellner & Dettner, 1996; Tabadkani & Nozari, 2014). Although P. fuscipes adults have been found in the stomach contents of frogs in Japan (Kurosa, 1958), their defensive effectiveness against vertebrate predators has not been experimentally tested before.

In this study, we used P. nigromaculatus as a vertebrate predator to assess the defensive efficacy of adult P. fuscipes. Although female beetles contain significantly more pederin than males (Kellner & Dettner, 1995), the predation rate on females was not markedly different from that on males (Table 2). Moreover, since P. fuscipes does not exhibit reflex bleeding (Dettner, 1987), the release of pederin requires injury. However, some frogs rejected P. fuscipes immediately—within 2 s—after tongue contact (Fig. 3A; Table 2). This rapid response suggests that compounds other than pederin, such as those secreted from abdominal glands, may also play a role in deterring pond frogs.

In addition to chemical defenses, P. fuscipes adults display a conspicuous reddish-orange and black body coloration, which may function as an aposematic signal (Tabadkani & Nozari, 2014). This color pattern resembles that of S. flavipes nymphs, which co-occur in the same habitats (Hayashi, 2023). In the present study, using P. nigromaculatus as a shared predator, we tested the potential mimetic relationship between P. fuscipes adults and S. flavipes nymphs. Frogs previously exposed to S. flavipes nymphs showed reduced attack and predation rates on P. fuscipes (Fig. 4A), suggesting that aversive learning from encounters with S. flavipes nymphs reduces the motivation to attack P. fuscipes. Conversely, prior exposure to P. fuscipes slightly increased predation on S. flavipes nymphs (Fig. 4B), although the effect was not statistically significant (Table 6). Thus, prior experience with P. fuscipes may not substantially alter predatory motivation in pond frogs.

Assassin bug nymphs as mimics

Many assassin bugs deter predators by injecting saliva or venom with their proboscises (Eisner, Eisner & Siegler, 2005; Schmidt, 2009; Walker et al., 2016), and also utilize scent gland secretions as chemical defenses (Louis, 1974; Staddon, 1979). In adults of S. flavipes, both stabbing and chemical defenses are effective against P. nigromaculatus (Sugiura & Hayashi, 2023). However, the anti-predator strategies of assassin bug nymphs remain largely unexplored. In this study, we assessed the defensive effectiveness of S. flavipes nymphs against pond frogs. Frogs often rejected nymphs immediately after tongue contact (Figs. 3 and 4), suggesting that surface chemicals act as effective deterrents. Unlike adults, which can stab predators, nymphs likely rely primarily on chemical defense.

Some adult assassin bug species share conspicuous coloration with other insects, such as wasps, that inhabit the same environments (Maldonado Capriles & Lozada Robles, 1992; Zhang & Weirauch, 2014; Alvarez, Zamudio & Melo, 2019), suggesting the presence of mimicry rings. While adult S. flavipes exhibit yellow and black coloration, nymphs display a reddish-orange and black pattern similar to that of P. fuscipes adults (Fig. 5). In a previous study, Sugiura & Hayashi (2023) demonstrated quasi-Batesian mimicry between adult S. flavipes and the bombardier beetle Pheropsophus occipitalis jessoensis Morawitz, 1862 (Coleoptera: Carabidae), using P. nigromaculatus as a shared predator. In this study, we applied the same approach to examine the mimetic relationship between S. flavipes nymphs and adult P. fuscipes. Although both species were sometimes rejected by pond frogs, S. flavipes nymphs showed stronger defense than P. fuscipes adults (Fig. 4), potentially due to differences in body mass (Table 1), which is known to influence predation outcomes in pond frogs (Sugiura, 2018). Our results also indicate an asymmetry in mimetic benefits and costs: P. fuscipes adults appear to gain a protective advantage from mimicry, whereas S. flavipes nymphs may incur slight costs. This asymmetry is consistent with expectations for quasi-Batesian mimicry and suggests the potential for antagonistic dynamics between co-mimics. However, asymmetry in benefits and costs alone does not necessarily imply antagonism; the interaction may still retain mutualistic elements depending on the ecological context and the magnitude of fitness consequences.

Figure 5 Two mimicry rings associated with assassin bugs.

(A) Reddish-orange and black mimicry ring: mimetic interaction of assassin bug nymphs (Sirthenea flavipes) with adult rove beetles (Paederus fuscipes). (B) Yellow and black mimicry ring: mimetic interaction of adult S. flavipes with adult bombardier beetles (Pheropsophus occipitalis jessoensis). The images of an adult S. flavipes and an adult P. occipitalis jessoensis were modified from Sugiura & Hayashi (2023). Photo credit: Shinji Sugiura.

Mimicry rings

A mimicry ring consists of at least two Müllerian co-mimics or one aposematic species plus one Batesian mimic (Kunte, Kizhakke & Nawge, 2021). There are diverse mimicry rings, each associated with a distinct aposematic color pattern, ranging in size from a minimum of two species to over 100 species at their largest (Wilson et al., 2015; Pekár et al., 2017; Quicke, 2017; Motyka, Kampova & Bocak, 2018; Motyka et al., 2021; Kunte, Kizhakke & Nawge, 2021; Leavey et al., 2021; Chatelain et al., 2023; Perger, 2024; Van Dam et al., 2024). The evolution and stability of these rings are shaped by both positive and negative frequency-dependent selection (Kunte, Kizhakke & Nawge, 2021). Positive frequency dependence, characteristic of Müllerian mimicry, promotes convergence by increasing the survival of individuals sharing common warning signals. Conversely, negative frequency-dependent selection may favor rarity and maintain signal diversity, particularly in Batesian mimicry systems or polymorphic rings.

The assassin bug S. flavipes displays distinct body color patterns in its nymphal and adult stages: a reddish-orange and black pattern in nymphs (Fig. 5A) and a yellow and black pattern in adults (Fig. 5B) (Sugiura & Hayashi, 2023). These color patterns are mimetically associated with adult rove beetles (P. fuscipes) and adult bombardier beetles (P. occipitalis jessoensis), respectively, which coexist with S. flavipes in the same habitats in Japan (Figs. 5 and 6) (Hayashi, 2023; Sugiura & Hayashi, 2023). These aposematic color patterns are also commonly found in other insects such as lycid beetles and stinging wasps (Wilson et al., 2015; Quicke, 2017; Motyka, Kampova & Bocak, 2018; Motyka et al., 2021; Chatelain et al., 2023).

Figure 6 An adult rove beetle and an adult bombardier beetle as mimetic partners of assassin bug nymphs and adults.

An adult rove beetle, Paederus fuscipes (left), is much smaller than an adult bombardier beetle, Pheropsophus occipitalis jessoensis (right). Photo credit: Shinji Sugiura.

In S. flavipes, both nymphs and adults prey on mole crickets and share the same habitats (Hayashi, 2023). The question arises: why does S. flavipes have different aposematic color patterns at different life stages? We hypothesize that the evolution of distinct aposematic color patterns in S. flavipes may be driven by the presence of mimetic partner species corresponding to each life stage’s body size. If predators generalize not only based on color but also body size, both traits may play an important role in the evolution of mimicry. Even if S. flavipes displayed the same reddish-orange and black coloration in both nymphs and adults, predators would likely distinguish between the two due to differences in body size, especially when comparing S. flavipes nymphs with adult P. occipitalis jessoensis. The absence or rarity of mimetic partners with similar body size may cause the shift in aposematic color patterns across different developmental stages.

Notably, the insects with a reddish-orange and black color pattern and a similar body size to S. flavipes adults were not found in our study sites. However, P. occipitalis jessoensis, the mimetic partner of S. flavipes adults in Japan, is absent from South and West Asia, where S. flavipes is found. Interestingly, the adult body color pattern of S. flavipes in these regions is different from that in Japan. The head and pronotum of South and West Asian populations are more reddish than those of the Japanese populations, although the black and yellow pattern on other body parts is consistent across all populations (Chłond, Bugaj-Nawrocka & Sawka-Gądek, 2019). Interestingly, the adult body color pattern of S. flavipes in South and West Asia resembles that of another bombardier beetle, Pheropsophus (Stenaptinus) catoirei (Dejean, 1825), which shares the same distribution range (South and West Asia) (Chłond, Bugaj-Nawrocka & Sawka-Gądek, 2019; Fedorenko, 2021). The adult P. catoirei exhibits a reddish head and pronotum, with the same black and yellow pattern on other body parts (Fedorenko, 2021). Therefore, it is likely that adult S. flavipes may belong to different mimicry rings in South–West Asia and other regions.

Our findings suggest that S. flavipes may not be involved in simple bilateral mimicry alone, but rather participate in broader mimicry rings composed of multiple aposematic species sharing similar color patterns. The color patterns of S. flavipes, which resemble those of various beetles and hymenopterans (Wilson et al., 2015; Quicke, 2017; Motyka, Kampova & Bocak, 2018; Motyka et al., 2021; Chatelain et al., 2023), may thus be reinforced not only by specific mimetic partners such as P. fuscipes and P. occipitalis jessoensis, but also by co-occurring species with convergent aposematic signals. Incorporating a community-level perspective on mimicry provides a more complete understanding of the evolutionary dynamics shaping ontogenetic and geographic variation in warning coloration. Furthermore, ontogenetic shifts in mimicry, as seen in S. flavipes, may contribute to the diversification and modularity of mimicry systems, allowing different life stages to integrate into distinct mimicry rings and thereby facilitating evolutionary transitions in signal design across developmental boundaries.

Limitations of this study

This study has several limitations that should be acknowledged. First, our generalization experiments involved a short temporal interval (approximately 6 min) between exposures to different prey types. Such a short delay may not fully capture the duration over which predator learning and memory generalization occur in natural settings. This temporal proximity could either exaggerate or underestimate the degree of generalization, potentially affecting the interpretation of mimicry efficacy. Second, the behavioral assays were conducted using a single predator species, P. nigromaculatus. While this species is ecologically relevant, relying on a single predator limits the ability to generalize findings across broader predator communities, which are often diverse and exert varying selection pressures. These constraints should be considered when interpreting the ecological and evolutionary implications of the observed mimicry dynamics.

Conclusions

The assassin bug S. flavipes displays distinct aposematic body color patterns in its nymphal and adult stages (Fig. 5). Our results indicate that S. flavipes nymphs and adults engage in mimetic interactions with adult P. fuscipes and P. occipitalis jessoensis, respectively, which co-occur in the same habitat in Japan (Figs. 5 and 6). The pond frog P. nigromaculatus serves as a shared predator of both the nymphs and adults of S. flavipes (Fig. 5). However, the specific frog individuals acting as predators may differ between life stages. Smaller juvenile P. nigromaculatus are more likely to target S. flavipes nymphs than adults, whereas larger adult P. nigromaculatus are more likely to attack adult S. flavipes. Thus, while the predator species remains the same, the actual individuals attacking S. flavipes may differ between life stages. Such stage-specific predation dynamics suggest that ontogenetic shifts in aposematic signals may reflect differences in the size and sensory capabilities of predators encountered at each life stage (e.g., Grant, 2007).

Furthermore, mimetic relationships are not exclusively maintained by a single predator species but can be influenced by multiple predator species (Pekár et al., 2017; Postema, Lippey & Armstrong-Ingram, 2023; Sugiura & Hayashi, 2025), which may drive the formation and persistence of mimicry, as well as the diversification and stabilization of aposematic coloration (Postema, Lippey & Armstrong-Ingram, 2023). In particular, changes in predator identity associated with increasing body size during development could mediate the effectiveness of warning signals, leading to divergent selection pressures between life stages (Grant, 2007).

Differences in aposematic color patterns between immature and adult stages have been reported in other insects (e.g., Willmott, Elias & Sourakov, 2011; Medina et al., 2020). This phenomenon appears to contradict the prevailing theory (Medina, Wallenius & Head, 2020; Medina et al., 2020), which suggests that aposematic coloration should exhibit minimal variation, as consistent warning signals are more easily learned and avoided by predators (Joron & Mallet, 1998). However, ontogenetic color change may be more widespread and adaptive than previously assumed, particularly when early and late life stages differ in size, behavior, or predator identity (Booth, 1990; Medina, Wallenius & Head, 2020; Medina et al., 2020). In this study, we propose that the presence of mimetic partner species matching the body size of each developmental stage may be an important factor influencing the evolution of distinct aposematic color patterns. Nonetheless, developmental constraints and ecological divergence across life stages should also be considered in future studies. Further research is needed to test this hypothesis.

Supplemental Information

Supplemental Information 1 A pond frog rejecting an adult rove beetle

The frog (Pelophylax nigromaculatus) stopped attacking the adult rove beetle (Paederus fuscipes) immediately after its tongue contacted the beetle. Video credit: Shinji Sugiura.

Supplemental Information 2 A pond frog rejecting an assassin bug nymph

The frog (Pelophylax nigromaculatus) stopped attacking the assassin bug nymph (Sirthenea flavipes) immediately after its tongue contacted the assassin bug. Video credit: Shinji Sugiura.

Supplemental Information 3 Raw data

Supplemental Information 4 The ARRIVE guidelines 2.0: author checklist

We would like to thank H. Yamashita, H. Uchida, T. Daijima, and other members of the Laboratory of Insect Biodiversity and Ecosystem Science, Graduate School of Agricultural Science, Kobe University, for their assistance with the maintenance of insects and frogs. We also acknowledge the use of the artificial intelligence chatbot, ChatGPT-4o, for manuscript proofreading.

Additional Information and Declarations

Competing Interests

Author Contributions

Animal Ethics

Data Availability

The authors declare there are no competing interests.

Shinji Sugiura conceived and designed the experiments, performed the experiments, analyzed the data, prepared figures and/or tables, authored or reviewed drafts of the article, collecting insects and frogs used in this study, and approved the final draft.

Masakazu Hayashi conceived and designed the experiments, performed the experiments, authored or reviewed drafts of the article, collecting insects used in this study, and approved the final draft.

The following information was supplied relating to ethical approvals (i.e., approving body and any reference numbers):

The experiments were performed in accordance with the Kobe University Animal Experimentation Regulations (Kobe University’s Animal Care and Use Committee, No. 30–01, 2023–03).

The following information was supplied regarding data availability:

Data is available at Figshare:

Sugiura, Shinji; Hayashi, Masakazu (2025). Data from: Mimicry between adult rove beetles and assassin bug nymphs with unequal defenses: antagonistic or mutualistic?. figshare. Dataset. https://doi.org/10.6084/m9.figshare.28703744.

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
