# Peer review of "Mimicry between adult rove beetles and assassin bug nymphs with unequal defenses: antagonistic or mutualistic?"

_PeerJ, doi:10.7717/peerj.19942_

## Round 0.1 · original submission · Major Revisions

Dear Dr. Sugiura, I ask you to carefully analyze the reviewers' comments and supplement the manuscript with new experiments. You must be sure that the conclusions clearly correspond to the experimental results. I hope that the new version of your manuscript will be approved by the reviewers for publication.

·

Basic reporting

The topic of the manuscript under review is relevant and interesting for general ecologists. Mimicry among insects is a phenomenon of evolutionary adaptation that increases the chances of survival of individuals. The significant volume of the "Discussion" section indicates that the authors have thoroughly familiarized themselves with the scientific literature on the topic of the article. The results and conclusions are supported by statistical analysis. Despite this, this article contains some shortcomings and technical comments.

Experimental design

In the abstract, the first four sentences (lines 14–20) should be moved to the "Introduction" section and support the statements with literary citations. The text in the abstract should describe only the research results.
In the abstract and the text of the manuscript, at the first mention of an animal organism, in addition to the full Latin name, it is necessary to indicate the author's surname and the year of description of the species in accordance with the International Code of Zoological Nomenclature (lines 21, 22, 23, 73, 77, 90, 139, 141, 347, 390).
There is an incorrect abbreviation of the Latin names of species throughout the text (Pa. fuscipes, Pe. nigromaculatus). It is necessary to correct it to P. fuscipes, P. nigromaculatus.
I recommend moving lines 103–109, 119–125, 133–136 from the Materials and Methods section to the Introduction section. It is more appropriate to place a description of the biology and ecology of the species in the Introduction.
References to figures and literary citations should be separated by separate brackets (lines 344, 367).
Describe the limitations of the research and add it to the Discussion section.

Validity of the findings

I recommend moving lines 352–354 from the Discussion section to the Conclusions section.
References to literary sources are not allowed in the Conclusions section. Therefore, it is necessary to move lines 407–418 to the Discussion section.
I recommend removing links to figures from the Conclusions section.
This study has an insufficient sample to establish statistically reliable conclusions.

Additional comments

No comments.

·

Basic reporting

1. Inconsistent Framing of Quasi-Batesian Mimicry
Quasi-Batesian mimicry is presented as a type rather than a continuum within Müllerian mimicry involving unequal defense. Authors should clarify that quasi-Batesian outcomes depend on predator learning behavior and signal similarity, not just defense level. Suggested fix: Introduce the mimicry continuum concept and predator sampling strategies (e.g., Aubier et al. 2017; Sherratt 2008).

2. Lack of conceptual framework for asymmetry
Results showed asymmetric benefit, but authors equate this with antagonism without discussing degree of cost to the model. Current literature suggests weak asymmetry does not imply antagonism unless cost is substantial. Suggested fix: Interpret asymmetry in light of potential negligible fitness cost to the model species.

3. Missing Link Between Predator Generalization and Mimicry Theory
Generalization tests are described behaviorally but not tied back to mimicry theory. Predator generalization is central to signal convergence and mimicry stability. Suggested fix: Connect observed generalization to signal template formation (Ruxton et al., 2008).

4. Ontogenetic color shift framed without supporting context
Speculation on body-size-matching in ontogenetic color change lacks developmental or ecological context. No evidence cited for stage-specific predation or signal utility. Suggested fix: Cite studies showing ontogenetic aposematism (e.g., Medina et al. 2020) and mention habitat or predator differences.

5. Oversimplification of Mimicry Ring Structure
Authors focused on bilateral mimicry but mimicry rings involve multiple species reinforcing a signal. They did not discuss community-level stability of mimicry. Suggested fix: Briefly mention broader mimicry ring structure and frequency-dependent selection (Kunte et al., 2021).

6. Minor issues in sentence flow could be polished (e.g., the abstract lines 28–31 feel wordy and could be more direct).

Experimental design

1. Clarify how inter-individual differences in frog behavior (e.g., learning/memory variation) were accounted for beyond GLMM random effects.
2. Temporal generalization: The 6-minute delay between exposures is shorter than many predator learning durations. While authors acknowledged this (line 181), a discussion of how this might limit generalization or mimicry assessment is warranted.
3. Line 228: Randomization details: Clarify procedures to prevent sequence/order bias.
4. Lines 185–186: Some insects reused in Experiment II, but the potential confounding effects (e.g. insect fatigue, memory of predator) are not controlled or discussed. Consider including reuse as a random effect or covariate in the GLMM, or justify why reuse wouldn't bias outcomes.
5. Lines 169–182: Authors say frogs were previously used in Experiment I, then again in Experiment II 6 minutes later. There is no discussion of how this prior exposure may affect their behavior beyond “memory duration is short.” Indicate literature support for 6–7 minute intervals being sufficient to test generalization cleanly.
6. Lines 210–234: Authors said they used GLM/GLMM but did not mention about test for multicollinearity or overfitting, nor check residuals.
7. Line 184: Listed sex of insects but did not explore or control for sex in analysis. This matters since Pa. fuscipes males and females may differ in chemical defense. If not, provide evidence.
8. Line 181: Mentioned “memory durations reported for other predators” – cited Ráčka et al. and Kojima & Yamamoto but gives no concrete durations. Add actual reported durations and clarify predator species for better context.

Validity of the findings

1. Table 5: Interaction effect (species × encounter) is not significant (p = 0.1265), but conclusions still emphasize a strong asymmetry. Authors should temper claims of interaction unless backed by significance or justify interpretation using effect size.
2. Tables 3–6: Reported means and odds ratios lack confidence intervals, which are required per ARRIVE and PeerJ standards. Add 95% CIs to odds ratios and marginal means in both tables and text.

Additional comments

1. Line 183: “cf. Sugiura & Sato, 2018” – not all readers understand “cf.” usage.
2. Broader implications: Expand briefly in Discussion on the evolutionary implications of ontogenetic mimicry shift in S. flavipes.

---

## Round 0.2 · Minor Revisions

Dear Dr. Sugiura, I ask you to make some minor additions and corrections to the manuscript before the article is accepted for publication.

·

Basic reporting

The authors of the article took my comments into account. They made changes to the text. The manuscript can be recommended for publication.

Experimental design

The authors of the article took my comments into account. They made changes to the text. The manuscript can be recommended for publication.

Validity of the findings

The authors of the article took my comments into account. They made changes to the text. The manuscript can be recommended for publication.

Reviewer 3 ·

Basic reporting

The present manuscript addresses a fascinating biological phenomenon (mimicry) which is likely to be of broad interest to the biological community. The study is skilfully and engagingly written. The authors have succeeded in presenting complex ecological and evolutionary questions in a clear and accessible form, demonstrating a high level of scientific and communicative expertise.

As illustrated in Figure 23 of the PDF document, the nomenclature of the species of amphibian is erroneously inscribed without a space between the constituent elements. It is imperative that the taxonomic classification of the species under consideration is corrected to "Pelophylax nigromaculatus". It is imperative that all scientific names are italicised and correctly spaced throughout the text, including in the Abstract and Keywords.

Experimental design

The methodology is presented in a thorough and logical manner, ensuring the reproducibility of the study. The methodological approaches employed by the authors can be considered a potential standard for future research on mimicry systems.
The experimental design is both well-structured and meticulously implemented. The authors conducted behavioural assays under controlled laboratory conditions, using a relevant and ecologically appropriate predator species (Pelophylax nigromaculatus) that naturally co-occurs with the studied insects. The clear separation between initial response trials and generalization tests allows for a nuanced understanding of predator learning and signal generalization. The sample sizes are considered adequate in view of the constraints inherent to live animal experiments, and the authors have ensured that biological replication has been achieved across individuals. The experimental procedures employed, including acclimation periods, hunger standardization, and control for satiation, are methodologically sound and adhere to established practices in the field of behavioural ecology. The statistical analyses are appropriate and clearly described. The utilisation of both generalized linear models (GLMs) and generalized linear mixed models (GLMMs) is both substantiated and implemented with precision, with particular emphasis placed on model fit, overdispersion, and multicollinearity. The reporting of odds ratios and confidence intervals is conducive to the transparency of interpretation. The incorporation of estimated marginal means (EMMs) serves to enhance the clarity of interaction effects. The experimental methodology and statistical treatment of the data are of a high standard, providing a reliable basis for the study's conclusions.

Despite the authors' interpretation of predator responses as influenced by prey species identity (Table 2), their statistical analyses (Table 3A and 3B) fail to demonstrate significant effects of species or body size on predation success. One potential explanation for this phenomenon is the occurrence of model overfitting or multicollinearity among the covariates. In order to strengthen the inference, it is recommended that a model selection procedure based on the Akaike Information Criterion (AIC) be applied. To elaborate further: The construction of a set of candidate Generalised Linear Models (GLMs) should be initiated with a null model as a foundation, to which species identity, body size, frog size, and interaction terms should then be added in a stepwise manner. The second step involves the comparison of models using AIC, with the model that exhibits the lowest AIC being designated as the most parsimonious. In the event of the presence of overdispersion, the utilisation of quasi-AIC (QAIC) is advised. This approach will enable the authors to identify which variables most strongly explain variation in predation outcomes and may reveal support for species-specific effects that were not significant under the full model structure.

Validity of the findings

The authors successfully identified a key behavioural pattern, namely the significant influence of prior encounter history on frog predation responses (as supported by Generalised Linear Mixed Model analysis; see Table 5). This finding provides a robust foundation for addressing the central research question. It also serves as a unifying thread that connects the major conceptual sections of the manuscript, including the following: pond frogs as predators, adult rove beetles as mimics, assassin bug nymphs as mimics, and mimicry rings. The integration of these elements results in a coherent and well-structured narrative.
Of particular note is the authors' critical reflection on the study's limitations, including the restricted predator spectrum and the limited duration of the experimental intervals. This contributes to the credibility and transparency of their work. The conclusions are well substantiated by the experimental evidence and hold both practical and specialised scientific value. This study is noteworthy for its exemplary integration of rigorous behavioural ecology research with a highly engaging and accessible presentation, thereby contributing to the broader popularisation of scientific knowledge.

---

## Round 0.3 · accepted · Accept

Dear Dr. Sugiura, I congratulate you on the acceptance of this article for publication. I hope that this article will arouse considerable interest among readers and will be positively received by your colleagues.

Reviewer 3 ·

Basic reporting

The authors have implemented all the recommendations of the reviewer. The quality of the manuscript has been significantly improved. I recommend the article for publication.

Experimental design

The authors have implemented all the recommendations of the reviewer. The quality of the manuscript has been significantly improved. I recommend the article for publication.

Validity of the findings

The authors have implemented all the recommendations of the reviewer. The quality of the manuscript has been significantly improved. I recommend the article for publication.